# Learning Preconditions of Hybrid Force-Velocity Controllers for Contact-Rich Manipulation

**Jacky Liang**
Robotics Institute
Carnegie Mellon University
jackyliang@cmu.edu

**Xianyi Cheng**
Robotics Institute
Carnegie Mellon University
xianyic@cmu.edu

**Oliver Kroemer**
Robotics Institute
Carnegie Mellon University
okroemer@cmu.edu

**Abstract:** Robots need to manipulate objects in constrained environments like shelves and cabinets when assisting humans in everyday settings like homes and offices. These constraints make manipulation difficult by reducing grasp accessibility, so robots need to use non-prehensile strategies that leverage object-environment contacts to perform manipulation tasks. To tackle the challenge of planning and controlling contact-rich behaviors in such settings, this work uses Hybrid Force-Velocity Controllers (HFVCs) as the skill representation and plans skill sequences with learned preconditions. While HFVCs naturally enable robust and compliant contact-rich behaviors, solvers that synthesize them have traditionally relied on precise object models and closed-loop feedback on object pose, which are difficult to obtain in constrained environments due to occlusions. We first relax HFVCs' need for precise models and feedback with our HFVC synthesis framework, then learn a point-cloud-based precondition function to classify where HFVC executions will still be successful despite modeling inaccuracies. Finally, we use the learned precondition in a search-based task planner to complete contact-rich manipulation tasks in a shelf domain. Our method achieves a task success rate of 73.2%, outperforming the 51.5% achieved by a baseline without the learned precondition. While the precondition function is trained in simulation, it can also transfer to a real-world setup without further fine-tuning. See supplementary materials and videos at https://sites.google.com/view/constrained-manipulation/.

**Keywords:** Contact-Rich Manipulation, Hybrid Force-Velocity Controllers, Precondition Learning

## 1 Introduction

Robots operating in human environments, like homes and offices, need to manipulate objects in constrained environments like shelves and cabinets. These environments introduce challenges in both action and perception. Environmental constraints reduce grasp accessibility, so robots must use non-prehensile motions that leverage object-environment contacts to perform manipulation tasks. For example, a book placed in the corner of a shelf has no collision-free antipodal grasps, but a robot can retrieve the book by first pivoting or sliding the book to reveal a grasp. Environmental constraints also introduce occlusions both before and during robot-object interaction, making precise object modeling and closed-loop vision-based feedback impractical.

In this work, we tackle these challenges by first choosing hybrid force-velocity controllers (HFVCs) that use force control to maintain robot-object contacts and velocity control to achieve the desired motion. Prior works in HFVCs require accurate object models, object trajectories, known contact modes, and closed-loop feedback of object pose and contacts. Our work relaxes these requirements by allowing HFVC synthesis

6th Conference on Robot Learning (CoRL 2022), Auckland, New Zealand.

and execution in more realistic settings with incomplete models, partial observations, and without closed-loop object feedback. While HFVCs are naturally robust to small modeling errors and collisions, large model mismatches inherent in constrained manipulation may lead to unsuccessful motions. To address this, we learn a precondition function that predicts when an HFVC execution will or will not be successful. The precondition function is a neural network that takes as input segmented point clouds of the scene and the HFVC skill parameters. It is trained entirely in simulation, and using point clouds without color information enables easier sim-to-real transfer. It filters for successful HFVC actions, and we use it in an online search-based task planner to reactively plan sequences of HFVC and Pick-and-Place skills. In our experiments, our approach allows a robot to slide, topple, push, and pivot objects as needed to manipulate cuboid and cylindrical objects in an occluded shelf environment. See Figure 1 for an overview of the proposed approach.

The main contributions of our paper are: 1) A compliant manipulation skill through an HFVC synthesis framework that relaxes previous requirements on object and environment modeling. This skill can achieve diverse motions like pushing, pivoting, and sliding through generating different parameters. 2) A point-cloud based precondition function for this HFVC skill that predicts if an HFVC execution with the given parameter and current observations would be successful, as some may fail due to inaccurate modeling. 3) Using an search-based planner with the HFVC skill and a Pick-and-Place skill to complete contact-rich manipulation tasks in an occluded shelf domain.

## 2    Related Works

**Nonprehensile Manipulation.**

Nonprehensile manipulation focuses on controlling contact interactions and planning contact-rich motions. For control, if we know the object's geometric and dynamics models and relevant contacts, then traditional methods can synthesize controllers that perform behaviors like rolling and slipping [1, 2]. In order to precisely maintain or change the state of contacts, hybrid controllers have been developed based on reasoning about contact modes [3, 4, 5]. Compared with traditional methods require accurate models, hybrid and velocity controllers [6] are more robust and have been deployed in industry for decades for tasks such as polishing Recently, a hybrid force velocity control (HFVC) method [7] has been developed for general quasi-static contact-rich motion tasks under uncertainty. HFVC is shown to be robust to modeling errors and contact inaccuracies and can perform general constrained manipulation tasks.

Recent works plan multi-step nonprehensile manipulation by directly planning with contact modes with optimization [8, 9, 3, 10] or sampling based planning methods [11, 12]. These methods require accurate models and lack the ability to plan and optimize for longer horizons. If skills and transitions are predefined, high-level optimizers like [13] can generate the skill and transition sequence for a longer horizon. However, such optimizers are subject to local optima, often not able to find solutions without good initialization.

Learning approaches can relax modeling assumptions for nonprehensile manipulation. Some works learn dynamics models [14, 15], while others directly learn policies [16, 17, 18]. They also make additional assumptions on objects and environments, such as uniform objects and full state information, and they do not plan with different types of nonprehensile strategies. While there are works on learning-based skill sequencing [19, 20, 21, 22, 23], they do not address planning with prehensile skills and different nonprehensile skills in more realistic settings, where skill execution success is unreliable and cannot be directly optimized by the planner.

Compared with the previous works, our method integrates control, planning, and learning into one system that can perform contact-rich tasks in the real world. We relax modeling assumptions of HFVCs by learning precondition models that predict when skill executions will be successful despite modeling inaccuracies. We exploit the advantages of HFVCs in a search-based planner — its ability to represent diverse non-prehensile motions, robustness, and the reduced required computation for planning contact-rich manipulation.

**Pregrasp Manipulation.** Another area of related works is pregrasp manipulation, where a robot must perform nonprehensile motions not to directly manipulate an object but to enable future grasps. A common setting is grasping in clutter, where a target object must be first singulated before it can be grasped. Singulation is usually done with a pushing policy to maximize downstream grasp, and it can be hardcoded [24], planned [25], or learned [26, 27, 28, 29]. In singulation, objects that prevent access to a target object can

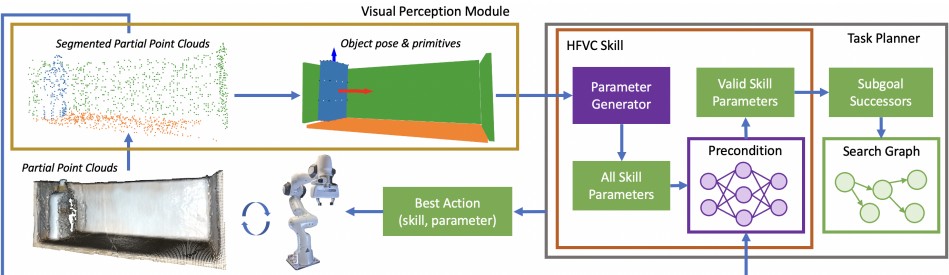

Figure 1: Overview of our approach that manipulates objects in constrained environments by planning with Hybrid Force-Velocity Controllers (HFVCs). Given partial point clouds, we first estimate object-environment segmentations and object and environment geometries. These are used to generate HFVC parameters, which are object subgoal poses and robot-object contact points. Due to model and feedback inaccuracies, not all generated parameters will lead to successful HFVC executions. As such, we learn the HFVC precondition, which predicts skill success from segmented point clouds and skill parameters. The planner uses the subgoal poses from the successful parameters to find the current best action, which is a (skill, parameter) tuple. The robot then executes this action, and replanning is done as needed.

be pushed away. In constrained environments, it is the unmovable environment, like tables or shelf walls, that prevent grasping. Under this context, many works study how to push an object on a table surface over the table edge to expose a grasp. Some works plan with known object models [30, 31, 32], learned dynamics [33], or explicit surface constraints [34], and some directly learn an RL policy [35, 36]. Recent works also proposed online adaptations of object dynamics, like mass and center of mass, to plan for table-edge grasps [37, 38]. While these works show how nonprehensile manipulation can enable downstream grasps, they typically cannot plan with multiple types of nonprehensile behaviors. Furthermore, they focus on revealing grasps, and not object manipulation in general, which sometimes may not require any grasps.

## 3 Method

**Problem Definition and Assumptions.** We tackle the problem of moving rigid objects from a start to a goal pose in constrained environments by a robot arm. The start and goal poses may be in different stable poses, and the object may not have any collision-free antipodal grasps at these poses. There is only one movable object in the environment. For actions, the robot arm can perform joint torque control. For perception, we assume access to an end-effector force-torque sensor and partial point clouds from an RGB-D camera, from which we can estimate segmentation masks and object poses. For skill parameter generation, we assume object geometries are similar to known geometric primitives, and that we can estimate these primitive shapes and environmental constraints from the segmented point clouds. Note this assumption does not apply to our learned precondition model and controller synthesis algorithm. Lastly, we assume object dynamic properties are within a reasonable range that enables nonprehensile manipulation by our robot arm.

**Approach Overview.** At the high level, the inputs to our system are partial point clouds that represent the current observation, and the outputs are actions represented as parameterized skills. Our approach has three main components. The first is synthesizing HFVCs that allow for compliant execution with inaccurate object models and feedback. The second is learning the HFVC skill precondition to filter out motions that will likely fail due to modeling mismatches. The third is using the learned precondition to plan sequences of skills for object manipulation tasks in constrained environments. The planner uses subgoals from the skill parameters as the skill-level transition model. The planner replans if the reached state deviates a lot from the subgoal. See Figure 1.

**Parameterized Skill Formulation.** We follow the options formulation of skills [39, 40]. We denote a parameterized skill as $o$ with parameters $\theta \in \Theta$. In our work, a parameterized skill $o$ has five elements: a parameter generator that generates both feasible and infeasible $\theta$, a precondition function that classifies skill success given current observations and skill parameters, a controller that performs the skill, a termination condition that tells when the skill should stop, and a skill-level dynamics model that predicts the next state after skill execution terminates. We assume the parameter generator, controller, and termination conditions are given. We assume skills have subgoal parameters, which contains information about the next world state if skill execution is successful. For example, an HFVC skill parameter will contain the

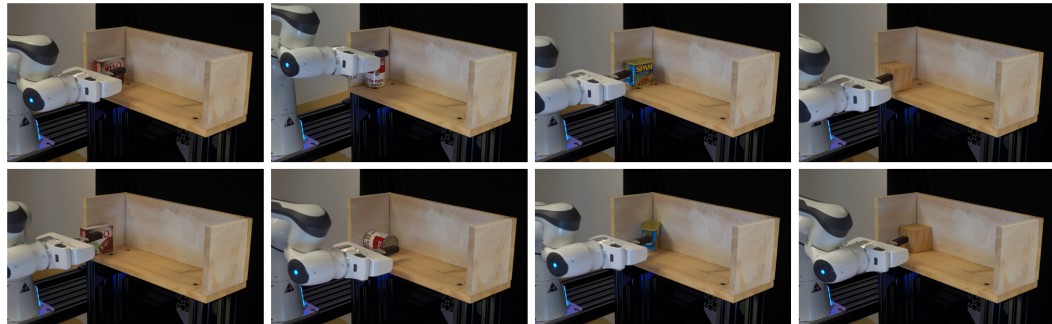

Figure 2: Our HFVC parameter generator gives diverse contact-rich behaviors. Each column is an HFVC skill execution. Top: initial states. Bottom: final states. Left to right: sliding, toppling, pivoting, and pushing.

desired object pose, and the planner assumes the object will reach the desired pose if 1) that parameter satisfies preconditions and 2) the HFVC commands are computed using that parameter.

### 3.1 Hybrid Force Velocity Controller Skill

The HFVC skill moves an object using a given skill parameter $\theta$, which contains the initial object pose, the desired object pose and the robot-object contact(s). Poses are 6D $\mathrm{SE}(3)$. To achieve the desired object motions, we use an optimization-based solver to output a sequence of hybrid force and velocity commands for the robot to follow. Below, we explain how parameters are generated for both precondition learning and planning, how HFVC commands are synthesized and executed from the parameter $\theta$, and how we learn the precondition function to classify successful parameters.

**HFVC Skill Parameter Generation.** The HFVC skill parameter contains the initial and subgoal object poses and the robot-object contact point(s). We generate two types of subgoal poses: ones that are in the same stable pose as the current pose, and ones that are in "neighboring" stable poses (i.e., toppled by $90°$). Initial robot-object contact points come from evenly spaced points on the surface of object primitives. We filter out contact points at which the robot would collide with the environment either for the initial or subgoal object poses. These intended initial contact points should be treated as skill parameters (like ones that parameterize grasps [41]), as they may not be the actual robot-object contact points achieved during execution, due to the end-effector geometry and inaccurate pose estimation. This parameter generation scheme can generate a diverse range of 3D behaviors such as pushing, sliding, pivoting, and toppling. See Figure 2.

We assume known geometric primitives (cuboids and cylinders) for parameter generation, but this does not significantly affect the focus of this work. One reason is that many real-world objects resemble cuboids and cylinders, especially in interacting with constrained environments like shelves and cabinets. Another is that this assumption is only made for parameter generation (the precondition directly takes in point clouds), and it can still generate a wide range of parameters with only some satisfying preconditions. As such, the object primitive assumption does not overly limit the types of behaviors the HFVC skill can achieve, and the parameter generator is useful even if the real object cannot be perfectly represented by the primitives.

**HFVC Synthesis from Subgoals.** Our controller synthesis algorithm generates hybrid force-velocity commands $\mathcal{H}$ for the robot to execute. Initially, the algorithm needs the current object pose ${}^{W}\mathcal{T}_{O} \in \mathrm{SE}(3)$, desired object pose ${}^{W}\mathcal{T}_{sg} \in \mathrm{SE}(3)$, initial estimated object-environment contacts, and initial robot-object contact point. An HFVC command $h \in \mathcal{H}$ include the velocity control directions $T_{v} \in \mathbb{R}^{n_{v} \times 6}$ and magnitude $\eta_{v} \in \mathbb{R}^{n_{v}}$, and the force control direction $T_{f} \in \mathbb{R}^{n_{f} \times 6}$ and magnitude $\eta_{f} \in \mathbb{R}^{n_{f}}$. Here, we adopt maximum velocity control where $n_{v} = 5$ and $n_{f} = 1$. During HFVC execution, HFVC commands are computed at a frequency of 20Hz, and this procedure only needs the current estimated object pose (see next section).

There are three steps in the HFVC synthesis algorithm. First, we use a quadratic program (QP) to optimize $T_{v}$ and $\eta_{v}$ that moves the object to ${}^{W}\mathcal{T}_{sg}$ under the contact mode constraints. Second, we compute the force control direction $T_{f}$, which is chosen to be as close as possible to the robot-object contact normal $n_{h}$ while being as orthogonal as possible to the desired hand velocity direction $v_{h}$: $T_{f} = \mathrm{argmin}_{T_{f}}(\|T_{f}v_{h}\| + \|T_{f} - n_{h}\|)$. Third, we solve for the force control magnitude $\eta_{f}$ by trying to maintain

a small amount of normal contact forces on every non-separating environment contact under the quasi-dynamic assumption. If the robot-object contact normal is parallel to the desired hand velocity direction, we only do velocity control ($n_f = 0, n_v = 6$), which results in a pushing motion. See details in Appendix.

**HFVC Execution with Partial Information.** Computing new HFVC commands during execution requires object pose feedback ($^W\mathcal{T}_O$) at about 10Hz, which cannot be directly obtained due to occlusions in constrained environments and delay in perception and state estimation. As such, we estimate $^W\mathcal{T}_O$ from robot proprioception, and we constrain HFVC velocity commands to prevent inaccuracies in this estimation to drastically alter execution behavior. See Figure 3.

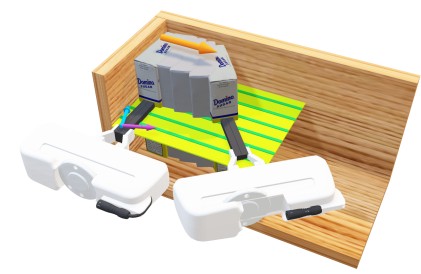

We first linearly interpolate a trajectory from the initial pose to the subgoal pose. Each point on the trajectory contains an object pose and its corresponding end-effector pose. To estimate the current object pose from robot proprioception, we assume static robot-object contacts (no slipping) during execution. Given the current end-effector pose, we find its closest end-effector pose in the interpolated trajectory. The corresponding object pose to the closest end-effector pose is the estimated current object pose.

HFVC is robust to pose estimation errors in the direction of the force controller, which moves the end-effector to maintain the desired robot-object contact. This is not the case for errors in the direction of the velocity controller, which will keep moving as if the object is still on the interpolated trajectory, leading to execution failure. To alleviate this issue, we constrain HFVC commands by projecting velocity control onto the plane that contains the interpolated end-effector trajectory, preventing the end-effector from traveling too far from the intended motion.

Figure 3: HFVC execution with partial information with example pivoting motion (start pose is object on the left). Gray objects represent linearly interpolated object poses used by proprioception-based object pose estimation. Object translation is exaggerated for visual clarity. Blue arrow represents direction of force command, purple is velocity. Velocity commands are projected onto the (green) plane containing the interpolated end-effector trajectory.

## 3.2 Learning HFVC Preconditions

Due to errors in visual perception (e.g. noisy point clouds), real-time feedback (noisy end-effector force-torque sensing, inaccurate object pose estimations), controls (e.g. HFVC solver does not take into account how the low-level controller actually achieves the commanded velocities and forces), and robot-object contacts (they are often non-sticking in practice), HFVC executions are not always successful. This motivates learning the HFVC skill precondition. The inputs to the precondition are segmented point clouds and the skill parameter. The output is whether or not executing the HFVC skill at the given state with the given parameter will be successful.

**Precondition Success Definition.** An HFVC skill execution is considered successful if it satisfies three conditions: 1) the object moved more than 1.5cm or 20° after skill execution, 2) the final pose is within 7cm and 60° of the subgoal pose (avoids pose differences that are more than an object's dimension away or in a different stable pose), and 3) the object does not move after the end-effector leaves contact. Since HFVC executions with model mismatches rarely reach exactly the intended subgoals, having a loose subgoal vicinity requirement allows the planner to execute more HFVC skills and make planning feasible. These thresholds are specific to a skill and indirectly depend on the object and environment properties. Tuning them will affect the positive-negative data ratio for training precondition models, but downstream task planning is somewhat robust to these changes, as the planner's prediction threshold for precondition satisfaction can be tuned after the model has been trained.

**HFVC Data.** The precondition is trained with HFVC execution data generated in simulation with cuboid and cylindrical-shaped objects from the YCB dataset [42]. To improve data diversity, we randomize object geometries by sampling non-uniform scales along object principal axes and randomly setting object mass and friction values. The range for both scaling and dynamics values are chosen so that the resulting object is feasible for manipulation in our shelf domain. See Figure 4 for visualization of the objects used. We also randomly sample the environment shelf dimensions, as well as the initial object pose and stable pose. From

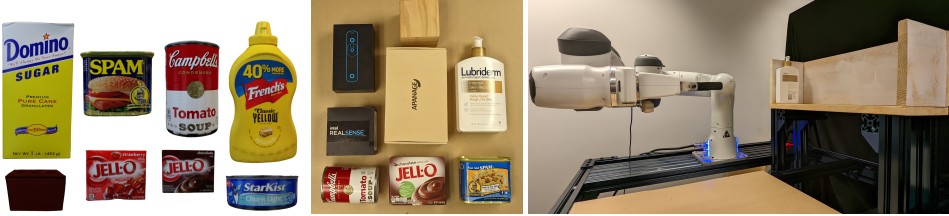

Figure 4: Object-in-Shelf Task Domain. Left: cuboid and cylindrical YCB objects used in simulation for training the HFVC precondition. Middle: objects used in real-world experiments. Right: real-world setup.

simulation, we obtain ground truth segmented point clouds and object poses after skill execution, the latter of which is used to compute ground truth skill preconditions. We use Nvidia IsaacGym, a GPU-accelerated robotics simulator [43, 44].

**Model Architecture.** The HFVC precondition is a neural network with a PointNet++ [45] backbone. The input point cloud is centered and cropped around the point that is in the center between the initial object position and the subgoal object position. This improves data efficiency as the network only has to reason about environment points relevant to object-environment interactions. The features of each point include its 3D coordinate as well as the segmentation label. For object points, we additionally append the skill parameter as additional features. These parameter dimensions are set to all 0's for the environment points. The PointNet++ backbone produces embeddings per point. We take the mean of the point-wise embeddings corresponding to the object points to produce a global object embedding. This embedding is passed to a multi-layer perceptron (MLP) to produce the final precondition prediction. The entire network is trained with a binary cross-entropy loss.

### 3.3 Search-Based Task Planning

Once the precondition is learned, we can use it along with the skill parameter generator to plan for tasks. We perform task planning in a search-based manner on a directed graph, where each node of the graph corresponds to a planning state, and each directed edge corresponds to a (skill, parameter) whose execution from the source state would result in the next state. In our domain, a task is specified by an initial and a goal object pose, and the planning state is the 6D object pose. For parameter generation, if the goal pose is close enough to the current pose, then it is included in the list of subgoal poses, so the planner can find plans that take the object directly to the goal pose. The output of the planner is a sequence of parameterized skills to be executed from the initial state.

We interleave planning graph construction with graph search, and search is performed in a best-first manner, similar to [46] which uses A* to perform task planning. However, it is difficult to efficiently perform A*-style optimal planning in our domain. This is due to several factors — 1) inaccurate transition models because we directly use subgoals as the next state, 2) expensive edge evaluation (node generation requires storing new point clouds for future precondition inference), and 3) high branching factor with many possible skill parameters at a given state. As such, we instead use Real-Time A* [47] with an inadmissible heuristic, where A* is performed until a search budget is exhausted or when a path to the goal is found. Then, the robot executes the first (skill, parameter) tuple of the path that reaches the best leaf node found so far. After skill execution, if the observed next state is not close enough to the expected next state, or we have reached the end of the current plan, we will replan again.

## 4    Experiments

Our experiments focus on evaluating our approach in a shelf domain, where a robot needs to move an object from a start pose to a goal pose. We perform three types of evaluations. The first gauges how much proprioception-based object pose estimation and velocity command projections improve HFVC execution success. The second is about HFVC skill precondition training performance. The third is on the overall task performance by running the planner with the learned HFVC skill precondition. A key comparison we make is between planning with and without the learned precondition function. Lastly, we demonstrate our planner and learned precondition on a real-world robot setup without further model fine-tuning.

|              | Skill Success | SG-ADD (cm)   | S-SG-ADD (cm) |
| ------------ | ------------- | ------------- | ------------- |
| Ours         | **52.3**%     | **6.9**±**8.1** | **3.4**±**1.9** |
| No-Feedback  | 38.6%         | 10.2±10.3     | 3.9±1.9       |
| No-Constraint| 46.0%         | 7.6±8.1       | 3.6±2.1       |
| No-Both      | 38.3%         | 10.3±10.3     | 3.8±1.9       |

Table 1: Skill execution evaluations for our approach vs. ablations that do not use proprioception-based object pose estimation and planar velocity constraints. We report skill success rate and error distance between the subgoal pose and the actual reached pose over all executions (SG-ADD) and over just the successful ones (S-SG-ADD). Numbers after ± are standard deviations.

|           | Ours | Full-PC | GT-Primitive | No-Params |
| --------- | ---- | ------- | ------------ | --------- |
| Accuracy  | 79%  | 76%     | **85**%      | 65%       |
| Precision | 78%  | 73%     | **85**%      | 60%       |
| Recall    | 77%  | 79%     | **83**%      | 79%       |

Table 2: Precondition training results with different input representations.

**Task Domain.** Our task domain consists of a 7 DoF Franka Panda robot arm, a rectangular shelf, and a set of objects for the robot to manipulate. Each task instance is specified by a different object geometry, shelf dimensions, shelf pose, initial object pose, and goal object pose. The robot successfully completes the task if it executes a sequence of (skill, parameter) tuples that brings the object from the initial pose to the goal pose, with a goal threshold of 1.5cm for translation and $10°$ for rotation. In addition to the HFVC skill, the planner also has access to a Pick-and-Place skill that can move objects via grasps if grasps are available.

### 4.1 HFVC Execution with Partial Information.

We first demonstrate the value of estimating object pose from proprioception and applying planar constraints on velocity commands. One ablation is running HFVC with "open-loop" pose estimation, where the current object pose is indexed from the interpolated trajectory with a time-based index (No-Feedback). This assumes the object is following the interpolated trajectory at a fixed speed. Another is running HFVC with proprioceptive feedback but without the velocity planar constraints (No-Constraint). The third variant is running HFVC without both of these modifications (No-Both). We report the execution success rate, the object's average discrepancy distance (ADD) between achieved and subgoal poses for all executions (SG-ADD) and only for the successful executions (S-SG-ADD). A total of 7.5k skill executions per method were used in these evaluations. See results in Table 1. Our approach achieves 52.3% success rate, higher than the ablations, with proprioception-based pose feedback (38.6%) making the most difference. These results demonstrate that both proprioceptive object pose feedback and planar velocity constraints can improve HFVC skill executions.

### 4.2 Precondition Learning

We generate the HFVC execution dataset with 6 cuboid-shaped and 2 cylindrical-shaped objects from the YCB dataset (see Figure 4). Train-test split is done across object scaling factors, with the test set having the smallest and biggest scales. To see how the network would perform with full, instead of partial observations, we include one ablation that uses full point clouds (Full-PC) and another that uses vertices of ground-truth object primitive meshes (GT-Primitive). We also train a variant that does not use skill parameter features (No-Params). See Table 2. Our method using partial point clouds is on par with Full-PC, and GT-Primitive performs the best. However, we cannot directly use GT-Primitive in planning due to having access to only partial point clouds. Instead, we include a comparison by using estimated object geometries in the task planning experiments below. No-Params performs the worst as expected, but it is still able to improve over random guessing due to biases in our collected data.

### 4.3 Task Planning Experiments

We evaluate task planning performance across several ablations in simulation. For each method, we run trials across the 8 YCB objects and 8 task scenarios, with 5 trials per object and scenario pair, resulting in a

|  | Ours | Est-Primitive | No-PC | Only-Pick-Place | No-Replan |
|---|---|---|---|---|---|
| Plan Success | **73.2**% | 61.1% | 51.5% | 27.4% | 28.0% |
| Plan Time (s) | 43.1±23.0 | 44.5±26.5 | 36.5±20.3 | 6.7±11.6 | 22.4±13.1 |
| Plan Length | 3.5±1.3 | 3.2±1.5 | 3.6±1.5 | 1.6±0.8 | 1.9±1.0 |

Table 3: Task performance of our approach using partial point clouds vs. using estimated object primitives (Est-Primitive), not using learned preconditions (No-PC), only using Pick-and-Place (Only-Pick-Place), and not doing replanning (No-Replan). Plan time includes replanning time. Numbers after ± are standard deviations. Plan time and length statistics are computed only over successful trials.

total of 240 trials per method. Each trial samples different initial and goal poses, and object geometries used in task evaluation are not in the training set. A task scenario specifies whether or not the initial object pose and goal pose are close to the shelf wall (4 variants) and whether they have the same stable pose (2 variants), for a total of 8 scenarios. We report the overall task success rate, average planning time and plan length for successful trials.

The first ablation is on using vertices from estimated object primitives (Est-Primitive) as the input to the precondition, instead of partial point clouds. The second is planning without the learned precondition, so the planner treats all generated parameters as feasible (No-PC). The third evaluates the usefulness of the HFVC skill in our domain by planning only with the Pick-and-Place skill (Only-Pick-Place). The last ablation does not replan and executes the entire found path from the initial state without feedback (No-Replan). For this method, we double the planning budget to 60s, so the planner is more likely to find a path all the way to the goal state.

See Table 3. The proposed approach achieves a success rate of 74.7%, higher than Est-Primitive (61.1%) and No-PC (48.9%), and much higher than Only-Pick-Place (22.9%) and No-Replan (28.8%). The drop in performance of Est-Primitive shows that while using ground truth primitives gives better precondition predictions, this improvement does not apply when primitives must be estimated from partial point clouds. The other ablations show the importance of learning HFVC preconditions, using HFVC skills in constrained environments, and replanning to compensate for inaccurate subgoal transition models.

**Real-world Demonstration.** Lastly, we demonstrate our planner with the learned precondition can operate in the real world. We use FrankaPy [48] to control the real Franka arm at 100Hz, similar to simulation. While the learned precondition can operate on real-world point clouds without further training, we had to tune low-level controller gains to reproduce similar contact-rich motions on the real robot. As with simulation, our planner is able to find plans that include a variety of contact-rich behaviors, like pushing, sliding, toppling, and pivoting, to manipulate objects on the shelf. Please see supplementary materials for more details and real-world videos.

**Failure Modes and Limitations.** The most common failure mode is precondition errors that lead the planner into finding infeasible plans or not finding a plan when there is one. This may be addressed by further improving precondition performance with more data, or enforcing domain-specific invariances. In addition, the object primitive assumption does prevent the parameter generator from supporting more complex objects, and this may be resolved by using learned parameter generator and visual perception module. For execution, our planner does not perform online precondition adaptation for unexpected object dynamics; doing so may improve task performance for objects with out-of-distribution dynamics parameters. Lastly, our method assumes only one movable object in the scene. Manipulating multiple objects may require learned perception systems and skill-level dynamics models that work in clutter.

## 5   Conclusion

HFVCs can naturally express Contact-rich manipulation behaviors. However, their reliance on precise models and closed-loop object feedback have prevented their use in cases where such information cannot be easily obtained. Our work 1) modifies HFVCs to not rely on privileged information and 2) learns where HFVCs are successful despite inaccurate models so 3) a planner can plan sequences of HFVC and Pick-and-Place skills to do contact-rich tasks in constrained environments.

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
