# OpenReview forum: "Learning Preconditions of Hybrid Force-Velocity Controllers for Contact-Rich Manipulation"
_robot-learning.org/CoRL/2022/Conference — CoRL 2022 Poster_

### Official Review · Reviewer_4YaQ · 2022-07-27

**Originality:** Good
**Technical Quality:** Good
**Clarity Of Presentation:** Good
**Impact:** 3

**Recommendation:**

Weak Accept: I recommend accepting the paper, but will not argue for my recommendation if the majority of other reviewers have a different opinion.

**Summary:**

This paper deals with the problem of manipulating objects under very constraining environmental conditions where non-prehensile skills are imperative to change the objects configuration so that prehensile manipulation is possible. To achieve this, the paper proposes to employ hybrid force-velocity controllers whose activation and sequencing are respectively driven by a learned precondition model and a classical search-based task planner. The proposed framework is tested to manipulate simple-geometry objects through pushing, pivoting, sliding and toppling skills.

**Issues:**

When improving this paper, the main points to be address are:

1. Position the paper more clearly w.r.t the state of the art, in terms of the use of HFVC skills, precondition models, and task planners.
2. Method description is unclear, incomplete and very hard to follow. Mathematical rigor is necessary when introducing variables, optimization problems, task planner and even the design of the neural networks.
3. A more specific discussion of the limitations of the proposed approach, targeting the specific contributions of this paper.

Please see my review on strengths and weaknesses above for an elaboration on the foregoing points.

**Quality Of The Limitations Section:**

Additional details required

**Reviewer Expertise:**

4: The reviewer is confident but not absolutely certain that the evaluation is correct

**Robotics Focus:**

Sufficient demonstration on hardware

**Strengths And Weaknesses:**

**Strengths**
1. The proposed approach works under more relaxed (realistic) conditions such as partial observations and no closed-loop feedback, making the proposed framework applicable under a broader range of settings.

2. The inclusion of a precondition model, which depends on the point cloud information, which allows the system to assess if the HFVC skill can be executed under the current environment conditions.

3. Performance of the proposed approach when compared to the ablation studies. The proposed framework is extensively evaluated against different ablation conditions, consistently providing better results under different settings.

**Weaknesses**
1. *Motivation and position of the paper w.r.t state of the art*:
  * When reading the motivation of this work, something that got my attention is the lack of related works. Aspects as simple as hybrid force-velocity controllers are not accompanied by a reference. More importantly, the full motivation of this work does not refer to any related work where non-prehensile manipulation strategies have been addressed in the past. This gives the impression that they ideas and motivation introduced in this paper are 100% new, which is very far from being true. As an example, one of the first works identifying and addressing the challenges of non-prehensile manipulation is [1], a PhD thesis from 1996.

    [1] K. M. Lynch, "Nonprehensile Robotic Manipulation:Controllability and Planning", CMU, PhD thesis. 1996.

  * In lines 59-60, the paper cites 7 papers on non-prehensile manipulation. However, my first concern about this is: why to cite that many papers without providing any insights about them? This, in my opinion, does not bring anything useful to position this paper w.r.t this bunch of 7 papers. Because of this, it is hard to see how useful these references are and how this paper positions w.r.t them.

  * The paper very briefly summarizes learning-based techniques in just a couple of lines (71-72), making special emphasis on their lack of considering multi-step tasks. However, this paper does not bring any contribution on this regard compared to different papers handling long-horizon tasks or skills sequencing, which was not reviewed in this paper. Therefore, I find this review of learning-based techniques a bit unfair, as it is unclear if the review focuses on the learning capabilities of the cited works or the multi-task features. If it is the latter, then the paper is missing to cite a large body of work addressing that specific problem.

2. *Method description*:
  * Lack of formalism: When introducing the proposed method, the paper disregard mathematical rigor, which is necessary in some parts of the method description. For example, it is not obvious how the objects/robot poses are represented, i.e., do we assume Euler angles, or rotation matrices, or quaternions? This detail is relevant, because depending on the orientation representation, one may need to consider specific constraints when working with different pose representations. In a similar line, there is no formal introduction of the hybrid force-velocity controllers and their interplay with the torque control framework assumed in the proposed method. Therefore, the lack of mathematical rigor leaves too many details open and makes the method description hard to understand in depth. Moreover, when introducing the search-based task planner, the definition of a planning state is unclear.

  * In lines 129-130, the paper describes that robot-object contact points are defined evenly spaced on the surface of the object primitives. However, it is unclear how the geometric shape of the robot end-effector may play a role here (if so). In other words, it seems that the robot-object contact points definition does not consider the geometry of the robot end-effector. If this is the case, this should be said explicitly, if not, then it is unclear how the geometry of the end-effector is considered to define this set of points.

  * In lines 147-155, the paper provides a very high-level (non-formal) description of a quadratic programming problem used to optimize the desired robot velocities. In the main text, there is no mathematical description of how this problem looks like, how it is formulated, and how it is solved. Equally important, this optimization problem seems to be the core of the framework when it comes to compute the desired velocity commands, but no limitations regarding the QP formulation and solution are discussed in the paper. Although the main text refers to the Appendix to know more details about the QP problem, I think an Appendix should serve as complementary contents and in my opinion, the QP formulation is very relevant to understand part of the proposed pipeline.

  * In lines 177-182, the paper reads: "... we project velocity controls onto the plane that contains the interpolated end-effector trajectory, preventing the end-effector from traveling too far from the intended motion. Note this is not the same as projecting onto the interpolated end-effector trajectory, which would overly constrain HFVC actions and may conflict with force controller commands." I read this several times and I still do not understand the motivation for the velocities projection on the plane of the robot's trajectory nor the reason(s) why the latter approach may not work.

  * Several method choices are not justified or motivated. For example, the values used for the precondition success definition are not discussed (do they work for any kind of robot manipulator? any kind of object geometry? etc). Moreover, the paper does not describe how these values were obtained. On a different note, the design of the model architecture is not discussed in detailed. For example, in lines 210-211, the paper describes that the skill parameters are added as additional features to the network, but no motivation is provided here. In summary, several model choices look very arbitrary.

  * Along similar lines to what I have pointed out above, the description of the task planner is very high-level, misses technical details and mathematical rigor. No definition of object/robot poses is given, no definition of planning state is provided, and no formal introduction of the real-time A* algorithm -- the one used in the paper ** is given.

3. *Limitations*:
  Lines 305-306 mentions some possible limitations coming from the task planner (which is not part of the contributions of this paper, in my opinion). However, the paper does not elaborate under which settings or conditions the task planner is not able to find a solution for tasks where a solution exists. On a different note, the paper does not discuss problems with the perception module, the most important contribution of this paper. Is it really flawless? Last but not least, no discussion about the QP formulation is provided, which is the part of the framework that outputs the desired velocity commands.

**Summary Of Recommendation:**

This paper provides a very high level description of the proposed method but lacks to provide enough technical details of the problem and designed framework, which makes very hard to assess the contributions and importance of this work. Moreover, the unclear position of this paper w.r.t the state of the art makes it even harder to understand the contributions of this paper. Because of this, the paper requires significant improvements in the method presentation, formalism and positioning w.r.t the state of the art.

-----

**Updated recommendation after rebuttal**

The revised version of the paper improved in several aspects: the updated related work section does a better job to position the paper w.r.t related works and makes more clear the contributions of this paper; The method description is introduced more formally and the added technical details allow the reader to better understand the technicalities of the proposed framework.

Most of my main concerns were satisfactorily addressed in the rebuttal and revised version of the paper. Therefore I updated my score accordingly.

---

> ### Author Response · Authors · 2022-08-22
> **Thank you for reviewing. Response Part 1**
>
> We thank the reviewer for the detailed and valuable comments, and we appreciate the time the reviewer has spent on evaluating our paper and providing this feedback. They will help us improve the paper and make the paper stronger. We will carefully address the reviewer’s concerns and make the suggested revisions in our new draft. We also thank the reviewer for recognizing how our approach allows applying HFVCs in more realistic settings, as well as the thoroughness of our experiments. Please see our responses to each of the concerns raised below. We also uploaded a new paper draft with our rebuttal responses with edits and additions colored in blue. Currently the paper does not satisfy page limits, but we want to make sure we address the reviewers’ concerns first before working to improve the formatting of the paper.
>
> **Motivation and related works.**
> The motivation for our work is to develop and study algorithms toward enabling contact-rich manipulation in constrained environments. We do this by building upon prior works, especially those that study Hybrid Force-Velocity Controllers for nonprehensile manipulation. In Related Works we discuss the different aspects of nonprehensile manipulation studied in the past, and we also state the novelty of our work in this context. In particular, our approach relaxes prior requirements on strict object modeling and feedback, and we can sequence multiple steps of different nonprehensile manipulation strategies to complete a task. We cited two recent works that synthesize HFVCs for robot manipulation (lines 63-64), and we have cited other papers by Professor Lynch on nonprehensile manipulation. We have added the reviewer’s suggested reference, since it is an earlier work and covers additional material relevant to this topic.
>
> > In lines 59-60, the paper cites 7 papers on non-prehensile manipulation....
>
> We agree with the reviewer that more clarity in positioning our paper with regards to prior work would improve the paper quality. We have made significant revisions to the related work section that covers nonprehensile manipulation, and we also added a new paragraph to better position our paper against prior works. Please let us know if the new content is sufficient. We will work on making the wording more concise to fit in the page limit later.
>
> > The paper very briefly summarizes learning-based techniques in just a couple of lines (71-72)...
>
> While our work performs task planning by sequencing skills, studying task planning is not the focus of our work, which is on enabling contact-rich manipulation. In the context of Related Works, lines 71-72 (now line 76) reference works that use model-free skill learning for nonprehensile manipulation, and we cite them as a counterpart to the previous paragraph that describes model-based approaches. This is not intended to be a review of learning-based approaches on task planning and skill sequencing. We have clarified this in the paper.
>
> **Lack of formalism.**
> We understand the reviewer’s frustration with the lack of mathematical symbols when describing our method. This was a choice we made during paper writing, as an earlier draft of the paper that included a full set of mathematical symbols and equations was too long, and we decided that moving other parts of the paper to the Appendix reduced the quality of the paper more so than moving the equations. Admittedly, we do rely on the Appendix for giving more detailed explanations of the method, and this is an unfortunate byproduct of the page limit. Our work, being systems-heavy and having many components, is especially impacted by the page limit. We brought some of the mathematical formulation back into the main paper, especially in the section discussing HFVCs.
>
> For object pose representation, our method uses 6D poses in SE(3), but they are stored and processed in different ways depending on the requirement of a particular computation. However, we only store and compute with quaternions and rotation matrices. For some operations it is more convenient to use one over the other, and we can convert these representations back and forth. We do not use Euler angles. We have clarified clarify this in the paper.
> With regards to the mathematical formulation of HFVCs, we added more formal notations in the main paper and also added a discussion on the interplay with the torque control framework in line 598. As with the new related works section, please let us know if this new material sufficiently addresses the raised concerns, and we will work on improving the paper formatting to fit everything in the page limit afterwards.
>
> The planning state for our experiments is the object pose, as stated in line 253. We have clarified that the pose is 6D.

---

> > ### Author Response · Authors · 2022-08-22
> > **Response Part 2**
> >
> > > In lines 129-130, the paper describes that robot-object contact points...
> >
> > In the context of skill parameter generation, robot-object contact points conceptually refer to the initial point on the object with which the robot should make contact; it is not the actual robot-object contact points achieved during skill execution, so it does not depend on end-effector geometry. Another way to interpret these contact points is by just treating them as some part of the parameters of a skill — different skill parameters will affect the outcome of the skill, and the planner/precondition models try to optimize these outcomes over different skill parameters, which include these initial robot-object contact points.
> >
> > > In lines 147-155, the paper provides a very high-level (non-formal) description of a quadratic programming problem...
> >
> > Please see the aforementioned additions to the HFVC formulations. The limitation of the modeling choice of our HFVC framework is discussed in the Appendix in Line 584 - 596. We did not include this in the main text primarily due to the page limit and also due to how QPs have been used in prior works to solve HFVCs, and it is not among the contributions of this paper.
> >
> > > In lines 177-182...
> >
> > We recognize this is a tricky concept that can be confusing to explain. First we will explain why projecting directly onto the interpolated end-effector trajectory does not work, and we will do so by referencing Figure 3. In Figure 3, a pivoting motion is desired. For the sake of this explanation, we can treat the final object pose in the Figure as the generated subgoal pose by the skill parameter generator. Then, the linearly interpolated object pose is shown in gray, and we can imagine a similar trajectory of end-effector poses sticking with those gray objects. If we project end-effector commands directly onto this interpolated end-effector trajectory, then the end-effector will likely lose contact with the object mid-execution, as the object will probably not slide so much and instead pivot around the corner of the shelf. If however, we merely project commands onto the plane, then the force controller will ensure robot-object contact is maintained. In general, if we override the force controller with position/velocity commands (which is what projecting onto a full 3D trajectory will do), we run the risk of either losing contact or pushing with too much force that impedes task progress.
> >
> > This illustrates how HFVC is robust to pose estimation errors in the direction of the force controller. However, this is not true in the direction of the velocity controller, so we constrain velocity commands to be near the interpolated trajectory to reduce compounding execution errors, so that even if there is large pose estimation error, the resultant robot trajectory would not be too far from the one that would’ve been induced by ground-truth pose information. The improvements from these constraints can be seen in Table 1, which reports skill execution success rates in simulation. We note that these improvements are more significant in the real world where the visual perception module performs worse and where object dynamics are more irregular.

---

> > > ### Author Response · Authors · 2022-08-22
> > > **Response Part 3**
> > >
> > > > Several method choices are not justified or motivated.
> > >
> > > For the precondition threshold values, they are skill-specific and depend on the skill’s performance, which implicitly depends on the end-effector and object geometry. We do observe a variation in skill success ratios across object geometries — see the top right plot of Figure 8 in the Appendix (page 7/12, labeled 17). The object with the lowest skill success rate is mustard bottle at around 40%, and the highest is the potted meat can at around 60%. If we make the precondition thresholds more or less strict, these success values would shift up and down. We chose these values to exclude reached poses that have likely rotated into a different stable pose or translated more than the object’s dimension away from the subgoal pose. However, we note that our method is flexible toward the specific thresholds used to label positive and negative precondition examples. This is because when the planner uses the learned precondition model, it can use a precondition prediction threshold other than 0.5, allowing us to lean toward less false positives or less false negatives, both of which have trade-offs for downstream task success. In general, these are hyperparameters that can be tuned to optimize downstream task planning success rate (it is not a good idea to tune these to optimize for precondition prediction accuracy, which may result in trivial labels for the supervised learning problem), but we did not do so, because training many precondition models and evaluating them in a full suite of planning experiments is costly in terms of both time and compute.
> > >
> > > The model architecture choices were made after many rounds of ablations and experiments. In the Appendix (section B.4, bottom of page 4/12, labeled 14), we provide a list of modifications we tried to apply to the model architecture and the training procedure in order to improve precondition learning performance. We arrived at the current architecture because these modifications did not improve learning results, and sometimes actually degraded performance. One modification we did not mention in the Appendix is using a completely different architecture, PerceiverIO (https://arxiv.org/abs/2107.14795), which is based on transformers. PerceiverIO had slightly faster learning but slightly worse test accuracy. What was interesting was that modifications to the training procedure seemed to affect both PointNet++ and PerceiverIO equally, and their relative performance rankings, even the performance gaps, never changed. Due to this, and due to the many ablations we performed on the model architecture, we believe that, at least in our domain, it is more likely for further improvements to the precondition prediction problem to come from improvements in data, and not from the model architecture.
> > >
> > > We have added clarifications to both of these points in the paper.
> > >
> > > > the description of the task planner is very high-level
> > >
> > > We chose not to include additional details regarding real-time A* in the paper due to three reasons: 1) it is well established in prior works and we do provide a reference to the original paper as well as other papers that use search-based planning for robot manipulation, 2) it is not the focus of our work, the content of which has been prioritized under the restrictions of 3) the page limit, which unfortunately made it impractical to cover both the breadths and depths of the many components of our proposed approach. We do agree that the clarity of the object pose and planning state representations could be improved in the paper, and we have made revisions accordingly.

---

> > > > ### Author Response · Authors · 2022-08-22
> > > > **Response Part 4 (final)**
> > > >
> > > > > Limitations: Lines 305-306...
> > > >
> > > > We clarify that the limitation in lines 305-306 (now 340-341) is not a limitation from the task planner, but rather from the inaccuracies of the precondition model. When the precondition model gives a false positive, the planner will generate a plan that is infeasible to execute. When the precondition model gives a false negative, the planner will ignore potential solution paths. This is the main failure mode and limitation of learning and planning with precondition models for performing downstream manipulation tasks, the main focus of our work.
> > > >
> > > > The visual perception module does have failure modes, the most important of which is that on noisy (real-world) point clouds, it may incorrectly predict the orientation or dimensions of object shape primitives, leading to generating infeasible skill parameters. This does not happen in simulation, but it does happen in the real world. We have included a more detailed discussion on the limitations of the visual perception module in the Appendix. We also clarify that the perception module is not among the contributions of this paper, which focus on learning and planning with preconditions of HFVC-based contact-rich manipulation skills.
> > > >
> > > > Please see the new details about the HFVC controller and the QP formulation we added in the main text. We do note that the more detailed formulation is still in the Appendix due to space constraints; this unfortunately is a challenge we are finding difficult to overcome without sacrificing other parts of the paper that are more unique to our main contributions.

---

> > > > > ### Comment · Reviewer_4YaQ · 2022-08-26
> > > > > **Reply to the authors about revised version**
> > > > >
> > > > > Dear authors,
> > > > >
> > > > > You have made a fantastic job in the revised version of the paper, thank you for taking the time to thoroughly answer to my comments, to clarify possible misunderstandings and apply some of my recommendations in the new version. I really appreciate the effort behind this.
> > > > > Now, concerning the introduced changes, the updated related work section does a better job to position the paper w.r.t the state of the art and makes more clear the contributions of this work. The method description is a bit more formal and the added technical details allow one to better understand the technicalities of the framework. Last but not least, I find the new details in the Appendix very useful.
> > > > >
> > > > > All in all, I think the paper improved after the revisions and I will change my score accordingly.
> > > > >
> > > > > Minor comments:
> > > > > 1. Notation
> > > > >  - Set of real numbers should be denoted by $\mathbb{R}$. (using \mathbb command)
> > > > > - Stick to a single notation for the special Euclidean group $\operatorname{SE}(3)$ (using \operatorname command)
> > > > > - In line 177: For the optimization problem, you may want to use: \DeclareMathOperator*{\argmin}{argmin}.
> > > > >
> > > > > 2. Figure 1 may provide a clearer high-level description if the data (variables) going from one block to another were included.

---

> > > > > > ### Author Response · Authors · 2022-08-26
> > > > > > **Thank you!**
> > > > > >
> > > > > > We appreciate the reviewer engaging with us throughout this process, and we are happy that our revisions were able to address your concerns. The new draft has also been updated with the new suggestions. Thank you for your valuable feedback and for raising the review score!

---

### Official Review · Reviewer_6oMV · 2022-07-30

**Originality:** Very Good
**Technical Quality:** Very Good
**Clarity Of Presentation:** Very Good
**Impact:** 4

**Recommendation:**

Strong Accept: I recommend accepting the paper and will argue for my recommendation even if other reviewers hold a different opinion.

**Summary:**

This work proposed a framework to solve contact-rich manipulation tasks, based on learning preconditions of HFVC. The method relaxed the requirements for accurate models and feedback of HFVC. The major contribution is learning the preconditions of HFVC to predict the outcome based on partial point clouds and skill parameters. It can be used for task planning, resulting in higher task performances. Furthermore, the learned model and framework can be transferred from simulated to real environments.

**Issues:**

As discussed in Weaknesses.

**Quality Of The Limitations Section:**

Limitations are addressed clearly

**Reviewer Expertise:**

3: The reviewer is fairly confident that the evaluation is correct

**Robotics Focus:**

Sufficient demonstration on hardware

**Strengths And Weaknesses:**

Strengths:
- It is a promising and meaningful topic to study the combination of nonprehensile and prehensile contact-rich manipulation in constrained environments. The paper proposed a great framework to approach it.
- The experiments and analysis are solid and thorough. The author compared different ablation studies to demonstrate the effectiveness of each component.
- The paper is well written and organized.
- The video greatly demonstrates the performance in action.

Weaknesses:
- The heuristic is inadmissible for the A* algorithm, but used for practical reasons. It should be mentioned in the main text.
- Line 96. It is a quite strong assumption that “we assume object geometries are similar to known geometric primitives”.
- Line 152. “the force control direction, which is chosen to be as close as possible to the robot-object contact normal while being as orthogonal as possible to the desired hand velocity direction.” Does it mean it is an average between the two orientations? What happens if they disagree?


**Summary Of Recommendation:**

The work introduced a framework to solve contact-rich manipulation in constrained environments. The experiments demonstrated the effectiveness of the learning preconditions of HFVS, and its application to search-based task planning. The paper is well written, and the experiments are well conducted.

---

> ### Author Response · Authors · 2022-08-22
> **Thank you for reviewing.**
>
> We thank the reviewer for the strong review and valuable feedback. We also thank the reviewer for recognizing the thoroughness of our experiments and how our precondition learning approach enables applying HFVCs to contact-rich manipulation. Please see our response to each of the weaknesses raised. We have uploaded a new paper draft with our rebuttal responses with edits and additions colored in blue. Currently the paper does not satisfy page limits, but we want to make sure we address the reviewers’ concerns first before working to improve the formatting of the paper.
>
> > The heuristic is inadmissible for the A* algorithm, but used for practical reasons. It should be mentioned in the main text.
>
> Thank you for pointing this out - we have added clarifications on this point in the paper when discussing the planning algorithm.
>
> > Line 96. It is a quite strong assumption that “we assume object geometries are similar to known geometric primitives”.
>
> Indeed this assumption in our work requires us to use only cuboids and cylinders in our experiments. However, the parts in our current pipeline that specifically rely on these shape primitives are the visual perception module (for pose tracking) and the skill parameter generator (for collision detection and generating delta poses, contact points, and grasps). There are ways this assumption can be relaxed. Using an off-the-shelf learned pose estimator, ideally one that does not require object-specific training, will be able to replace the current visual perception module and handle other object geometries. For the skill parameter generator, point-cloud collision detection can be done by a learned model as demonstrated in some prior works. Prior works have also demonstrated sampling 6 DoF grasps from point clouds, and similar techniques should also carry over to sampling contact points and delta poses. We did not pursue using these learned models to focus time and engineering effort toward the contributions of this paper, but we are confident that our method could be extended to more complex object geometries as the neither precondition model nor HFVC synthesis rely on these geometric primitives (the precondition model uses point clouds, and HFVC synthesis uses sparse contact points which can be inferred by learned methods).
>
> > Line 152. “the force control direction, which is chosen to be as close as possible to the robot-object contact normal while being as orthogonal as possible to the desired hand velocity direction.” Does it mean it is an average between the two orientations? What happens if they disagree?
>
> The force control direction is not the average between the two orientations but the result of an optimization problem whose objective contains terms relating to the two orientations. We've clarified exactly what this is in the Appendix line 565. If the two directions don’t agree, it physically means that the desired motion is pushing in the direction of the robot-object contact normal. In this situation, we will perform pure velocity control (see line 180) for the pushing motion. We have also clarified this point by adding a new mathematical explanation into the paper in line 177.

---

### Official Review · Reviewer_yUsy · 2022-08-01

**Originality:** Good
**Technical Quality:** Good
**Clarity Of Presentation:** Very Good
**Impact:** 3

**Recommendation:**

Weak Accept: I recommend accepting the paper, but will not argue for my recommendation if the majority of other reviewers have a different opinion.

**Summary:**

This manuscript uses hybrid force and velocity controllers(HFVC) as the skill representation and introduces a pipeline to plan skill sequences for contact-rich manipulation. The authors first modify HFVCs to work with inaccurate models and estimate object poses using the robot proprioception for closed-loop feedback. Then the authors propose to learn a point-cloud-based precondition classifier to indicate whether HFVC executions will be successful. Then those learned preconditions are used in a search-based task planner to complete contact-rich manipulation tasks. The authors apply the pipeline to manipulation tasks inside a shelf, achieving a task success rate of 73.2%, significantly outperforming a baseline without the learned precondition.

**Issues:**

1. In lines 163-170, it seems that the proposed object pose accuracy also depends on object geometry. It will strengthen the paper if more detailed analyses are provided, such as the errors of different objects.
2. In lines 192-193, “an HFVC skill execution is considered successful if it satisfies three conditions: 1) the object moved more than 1.5cm or 20◦ after skill execution, 2) the final pose is within 7cm and 60◦ of the subgoal pose” Are the performances sensitive to these parameters?
3. The learned preconditions are used for task planners. However, the learned precondition classifier is not 100% accurate. If the task requires several classification results, will the task success rate decrease significantly with respect to the number of precondition classification results?

**Quality Of The Limitations Section:**

Limitations are addressed clearly

**Reviewer Expertise:**

4: The reviewer is confident but not absolutely certain that the evaluation is correct

**Robotics Focus:**

Sufficient demonstration on hardware

**Strengths And Weaknesses:**

Strengths:
1. The paper is well written and easy to follow.
2. The paper introduces a framework to enable hybrid force-velocity controllers to accomplish various manipulation tasks without knowing the accurate model and object pose feedback.

Weaknesses:
1. There are many predefined rules in this work. Such as  the “prediction success definition” ( 1) the object moved more than 1.5cm or 20◦ after skill execution, 2) the final pose  is within 7cm and 60◦ of the subgoal pose) It’s not clear whether these rules are generally helpful in a more wide range of manipulation tasks. More questions about these rules are in the Issue section.


**Summary Of Recommendation:**

The manuscript introduces a framework to enable hybrid force-velocity controllers to produce compliant contact-rich behaviors to accomplish various manipulation tasks without knowing the accurate model and object pose feedback. The framework has a wide range of potential applications in the real world. However, there are multiple predefined rules in this work. It’s unclear whether these rules are generally helpful. Therefore, I recommend weak acceptance.

---

> ### Author Response · Authors · 2022-08-22
> **Thank you for reviewing.**
>
> We thank the reviewer for the helpful feedback which we will incorporate in our new draft of the paper. We also thank the reviewer for identifying how our framework allows HFVCs to perform nonprehensile manipulation skills under inaccurate object modeling, a key contribution of this work. In addition to adding clarifications to the paper, we have provided replies to each of the issues raised below. We also uploaded a new paper draft with our rebuttal responses with edits and additions colored in blue. Currently the paper does not satisfy page limits, but we want to make sure we address the reviewers’ concerns first before working to improve the formatting of the paper.
>
> **Issue 1.** Accuracy of the object pose estimator across different objects can be partially inferred from the top right plot in Figure 9 of the Appendix  (Page 8/12, labeled Page 18). This plot shows the distribution of ADDs between the final achieved object pose with the intended subgoal pose for skill executions that satisfied preconditions. ADD is the Average Discrepancy Distance between two poses of the same object computed by averaging the distances between corresponding object points. Since the proprioception-based object pose estimator outputs poses that are linearly interpolated between the initial object pose and the intended subgoal pose, and since that object pose estimation error tends to grow monotonically during skill execution, this plot can serve as an upper bound for the pose estimation errors. As the plot shows, there is no significant difference in pose estimation error distributions across different object types. We hypothesize that this is due to how the skill parameter generation scheme is similar across all objects regardless of their geometry.
>
> **Issue 2.** There are two criteria in question - one is for labeling whether or not a skill execution satisfied the precondition (7cm and 60 degrees), and one for labeling whether or not a planner execution completed the task (1.5cm and 20 degrees). For the task success threshold, the end user can set this value to an acceptable tolerance depending on the domain. This does not impact the workings of our algorithm, although it will impact its performance (e.g. a higher task success threshold will result in more successes). For the precondition threshold, these values are not task-specific but are rather skill-specific. However, we note that our method is flexible toward the specific thresholds used to label positive and negative precondition examples. This is because when the planner uses the learned precondition model, it can use a precondition prediction threshold other than 0.5, allowing us to lean toward less false positives or less false negatives, both of which have trade-offs for downstream task success. In general, these are hyperparameters that can be tuned to optimize downstream task success rate. We did not do so because of the high compute and time requirements (running full suites of planning evaluations and training many precondition models are expensive), but this can be done if needed to improve performance and to adapt the approach to a new domain.
>
> **Issue 3.** Currently the planning horizon is 3 steps, so the planner does need to sequence several precondition classification results, although the sequence is not very long, and most tasks in our domain can be completed within 5 steps. The comparison in Table 3 shows that without a learned precondition function (treating all skill parameters as valid) results in a task success rate of 51.5%, as opposed to 73.2% with the learned precondition function. Our current strategy is to plan relatively short-horizon plans quickly and perform replanning, as directly using subgoals as the skill-level transition model is not accurate enough for long-horizon planning, regardless of the precondition model’s performance. However, if we assume access to a perfect skill-level transition model, then with the ~80% classification accuracy our model achieves, we can roughly state that planning performance with the precondition model for problems with 8-step planning horizon may be degraded to the 51.5% of without the precondition model for problems with the 3-step planning horizon ($0.8^8 \approx 0.55^3$, where 55% is the prediction accuracy if one were to always predict positives). However, this estimate relies on many assumptions, such as the precondition model performance is uniform across all (state, parameter) tuples, so subsequent precondition predictions on a sequence of skill executions do not exhibit covariate shift. In practice however, we argue that due to inaccuracies in skill-level dynamics models, whether learned or not, replanning will likely yield better performance than directly performing long-horizon planning.

---

### Official Review · Reviewer_SLZA · 2022-08-01

**Originality:** Fair
**Technical Quality:** Good
**Clarity Of Presentation:** Good
**Impact:** 3

**Recommendation:**

Weak Accept: I recommend accepting the paper, but will not argue for my recommendation if the majority of other reviewers have a different opinion.

**Summary:**

This paper presents a way to integrate 3D perception in Hybrid Force Velocity Controllers (HFVC), where force control is used to maintain contact points between the robot and an object and velocity control is used to move the object, for example via a sliding motion. The paper proposes skills of HFVCs that can propose subgoals, and whose parameters are gated by a precondition module that tests when the skill is valid and can be applied. Skills incorporate input from 3D pointclouds, which can be used to get rough object segmentation, and their parameters include the desired object subgoal poses and robot-object contact points. Subgoals are then used in task planning, assuming knowledge of a skill transition model, and the paper evaluates this capability on a real robot with a limited set of simple rectangular or cylindrical objects.

**Issues:**

It would be great if the weaknesses W1-4 are addressed during the revision period. Aside from W4, they do not involve running new experiments, so this should hopefully be doable.

**Quality Of The Limitations Section:**

Limitations are addressed clearly

**Reviewer Expertise:**

4: The reviewer is confident but not absolutely certain that the evaluation is correct

**Robotics Focus:**

Sufficient demonstration on hardware

**Strengths And Weaknesses:**

The main strengths of this paper are (a) showing that compliant control can be achieved via feedback, even with noisy and inaccurate and incomplete object perception, and (b) that one can plan over skills as long as a skill transition model is provided. Additionally, the paper is generally well-written and the robot experiments are convincing.

The main weaknesses are:

W1: There doesn't seem to be much consideration of force/acceleration limits. In fact, the paper does not provide much detail about the formulation of the optimization problem used to solve HFVCs, given a desired object subgoal pose and robot-object contact points. I would have loved to have seen the mathematical formulation of this problem, currently placed in the Appendix, to be part of the main paper.

W2: Objects are assumed to be rectangular or cylindrical, which is limiting as there are multiple objects that do not fall under these categories (eg plates, cutlery, tools). It would be great to mention what changes would be required if a broader family of objects were allowed.

W3: Only a single object is assumed to be movable in the scene. This is sufficient for showcasing the controller, but does not test for object interaction in cluttered conditions. This is acknowledged in the limitations section, but it would be useful to describe what would need to change to handle multiple moving objects.

W4: It is not clear what the sim-to-real gap for the training of skill preconditioners in simulation is. If friction and contact are modelled poorly in the simulator, it seems like the preconditioner would also fail. It would be great to have a comparison of preconditioner errors when real vs simulated data is being used. It is not clear if Fig 10 in the Appendix refers to simulated data or real data. I suppose the former.


**Summary Of Recommendation:**

I am recommending a weak accept based on the fact that the proposed method is shown to be practically useful in non-prehensile manipulation, while using 3D perception. Although the perception system is trained separately from the sequence of HFVC skills, it is still an interesting solution to a challenging robotics problem, that I think the community would be happy to see. I cannot comment with certainty on the novelty of the method with respect to existing literature on HFVCs. If other reviewers suggested rejection, I would consider changing my recommendation.

---

> ### Author Response · Authors · 2022-08-22
> **Thank you for reviewing.**
>
> We thank the reviewer for the thoughtful review and feedback - they will help us make additional revisions and improve the paper. We also thank the reviewer for recognizing a main argument of our paper, which is that our proposed method allows us to use and plan over skills with inaccurate object modeling. Please see our response to each of the points raised in the weaknesses below. We also uploaded a new paper draft with our rebuttal responses with edits and additions colored in blue. Currently the paper does not satisfy page limits, but we want to make sure we address the reviewers’ concerns first before working to improve the formatting of the paper.
>
> **W1.** We understand the reviewer’s preference for including HFVC formulations in the main paper text. This was a choice we made in order to satisfy the page limit. We will try to bring more mathematical notation into the main text in our new drafts. As for the force/acceleration limits, our HFVC operates under the quasidynamic assumption — meaning that it is operated at a low speed. Force/acceleration limits of a robot arm are too high for our controller commands to hit, so we didn’t include this in our math formulation. We clarified this point in the Appendix. In practice, we also only experiment with objects that are within the robot workload, so that the force limit will not be hit.
>
> **W2.** The parts in our current pipeline that specifically rely on these shape primitives are the visual perception module (for pose tracking) and the skill parameter generator (for collision detection and generating delta poses, contact points, and grasps). Using an off-the-shelf learned pose estimator, ideally one that does not require object-specific training, will be able to replace the current visual perception module and handle other object geometries. For the skill parameter generator, point-cloud collision detection can be done by a learned model as demonstrated in some prior works. Prior works have also demonstrated sampling 6 DoF grasps from point clouds, and similar techniques should also carry over to sampling contact points and delta poses. We did not pursue using these learned models to focus time and engineering effort toward the contributions of this paper, but we are confident that our method could be extended to more complex object geometries as neither the precondition model nor HFVC synthesis rely on these geometric primitives (the precondition model uses point clouds, and HFVC synthesis uses sparse contact points which can be inferred by learned methods).
>
> **W3.** Multiple objects present two additional challenges - perception and dynamics predictions. Presence of clutter would reduce the amount of visible pixels each object has, so we expected degradations in many vision-dependent intermediate tasks like collision detection, sampling delta poses, contact points, and pose tracking. We also expect that directly using subgoal poses as the skill-level transition model will be too inaccurate to enable useful planning as doing so does not take into account object-object interactions, and learning a dynamics model would be even more challenging due to the increased occlusions. We do not have thorough answers to these questions at the moment and leave them to be addressed in future work.
>
> **W4.** We thank the reviewer for bringing up this point and agree that the sim-to-real gap of the precondition model is an important question. There are 3 sources of sim-to-real gap for the precondition model - the input point clouds, the generated parameters (the parameter generator depends on estimated object primitives and poses, which perform worse in the real world), and the interaction dynamics. We did randomize dynamics parameters (mass, friction) during simulation data generation to roughly cover the set of objects that our real robot, the Franka Panda arm, would be capable of manipulating (i.e. the robot has torque limits, so it cannot move objects that are too heavy or apply a lot of force for objects with very low frictions). We outfit the end-effector with a gripper extension covered in Plasti Dip to increase its friction (noted in the Appendix), which helps to make robot-object interactions more robust and reduce the sim-to-real gap. Anecdotally, we did not observe significant sim-to-real gaps for the precondition model due to dynamics mismatches. This is due to the robustness of our controllers and hardware as well as that the real-world objects do not have dynamics that significantly deviate from those of the objects in simulation. Most precondition failures in the real world came from the second source - inaccurate object perception leading to infeasible and out-of-distribution skill parameters that the model has not seen during training. We are trying to arrange for real-world evaluations of the precondition model at the moment, but this is challenging because the authors familiar with the real world setup are currently out-of-town on summer internships.

---

> > ### Author Response · Authors · 2022-08-24
> > **Update on W4**
> >
> > While it is challenging for us to collect new data due to authors being out of town, from our previous real-world execution logs we were able to recover some existing skill executions that could be used to evaluate the precondition model in the real world. We have updated these results in Appendix section B.5 (page 6/13, labeled 18), which we have copied below.
> >
> > To characterize the sim-to-real gap of the learned precondition function, we performed 61 skill executions across the real world objects in Figure 4 and recorded the prediction results in Table 5 (see below).
> > We show both the numbers for our approach and No-Params in simulation (from Table 2) and their performance on the real-world dataset (RW).
> > There is a 10% accuracy drop for our approach and a much larger 21% drop for No-Params.
> > Most of the accuracy drop came from reduced precision, meaning there are many samples where the model predicted were successful but in reality were not.
> > Recall actually increased for our approach, meaning a higher proportion of actual positives were predicted to be positive.
> > We note this is generally an acceptable trade-off for planning, as our replanning scheme allows recovering from false positive errors, but having high false negatives would prevent the planner from finding feasible solutions when they exist.
> > Lastly, we note that due to the degraded performance of the visual perception module on real-world data, the real-world precondition labels, which rely on real-world object pose estimates, are likely to be noisy, so we expect our models to actually perform better than these numbers would suggest.
> >
> > |           | Ours | Ours RW | No-Params | No-Params RW |
> > |-----------|------|---------|-----------|--------------|
> > | Accuracy  | 79%  | 69%     | 65%       | 44%          |
> > | Precision | 78%  | 53%     | 60%       | 35%          |
> > | Recall    | 77%  | 86%     | 79%       | 71%          |

---

> > > ### Comment · Reviewer_SLZA · 2022-08-25
> > > **Reviewer response**
> > >
> > > Thank you for addressing my concerns and for doing additional experiments. I will be recommending that this paper be accepted.

---

> > > > ### Author Response · Authors · 2022-08-26
> > > > **Thank you!**
> > > >
> > > > We appreciate the reviewer for the valuable feedback and suggestions, and we are happy our rebuttal and the new experiment helped address your concerns. Thanks again!

---

### Author Response · Authors · 2022-08-25
**New Paper Draft**

Please see the attached PDF for the new paper draft. The appendix and the main paper have been merged together for convenience. Edits and new additions are colored blue. Currently the paper does not satisfy page limits, but we want to address reviewers' concerns first and will work on formatting the paper later.

---

### Meta-Review · Area_Chair_DkMe · 2022-08-09

**Recommendation:** Accept (Poster)
**Confidence:** 4

**Metareview:**

Thank you for your submission to CoRL 2022. The reviewers have left comments below, with key comments summarized here. Please address these comments in your rebuttal.

**Strengths**
- The paper shows that compliant control can be achieved via noisy, inaccurate, and incomplete feedback
- It is a promising and meaningful topic to study the combination of nonprehensile and prehensile contact-rich manipulation in constrained environments.
- The paper is generally well-written
- The robot experiments are convincing. The proposed framework is extensively evaluated against different ablation conditions, consistently providing better results under different settings.

**Weaknesses**
- Objects are assumed to be rectangular or cylindrical, which is limiting as there are multiple objects that do not fall under these categories (eg plates, cutlery, tools). Please discuss what changes would be required if a broader family of objects were allowed.
- Only a single object is assumed to be movable in the scene. This is sufficient for showcasing the controller, but does not test for object interaction in cluttered conditions. What would need to change to handle multiple moving objects?
- It is not clear what the sim-to-real gap for the training of skill preconditioners in simulation is.
- There are several predefined rules in this work. Such as the “prediction success definition”. How much were these tuned for the capabilities of the method vs. what is needed for the real-world task?
- More discussion is needed of previous work on non-prehensile manipulation (see reviewer comments).
- There needs to be more mathematical rigor in the method description (see reviewer comments).

Post-rebuttal update:
The reviewers have discussed the authors' response and have concluded that most of the issues in the reviews were adequately addressed. They also agreed on the importance of the proposed approach for contact-rich tasks.

**Best Paper Nomination:**

No

---

> ### Author Response · Authors · 2022-08-22
> **Thank you for reiewing.**
>
> We thank the meta reviewer and all reviewers for their detailed and helpful feedback, as well as recognizing the novelty of our approach and strengths of our experiments. We will address reviewer feedback carefully both in our responses and in our new draft of the paper. We hope our rebuttal can help clarify the points raised by the reviewers, and we welcome additional comments and discussion. Please see below for a summary response to the weaknesses listed above. We also uploaded a new paper draft with our rebuttal responses with edits and additions colored in blue. Currently the paper does not satisfy page limits, but we want to make sure we address the reviewers’ concerns first before working to improve the formatting of the paper.
>
> **Known object primitives.**
> Two parts in our method rely on known object primitives - the visual perception module (for pose tracking) and the skill parameter generator (for collision detection and generating delta poses, contact points, and grasps). This assumption may be relaxed by employing learning-based approaches, for example by learning pose estimators, shape completion, and parameter generators. These capabilities have been demonstrated in prior works. We did not pursue using these learned models in order to focus time and engineering effort toward the contributions of this paper, but we are confident that our method could be extended to more complex object geometries as neither the precondition model nor HFVC synthesis rely on these geometric primitives.
> Single movable object.
> Extending our method to support multiple movable objects requires updating two parts of the pipeline - object perception and skill dynamics prediction. We expect that cluttered scenes will degrade the performance of intermediate vision tasks like collision detection, pose estimation, and shape completion. Dynamics prediction is also more challenging given complex object-object interactions. We do not have thorough answers to these questions at the moment and leave them to be addressed in future work.
> Sim-to-real gap.
> There are three sources of sim-to-real gap for the precondition model - sim-to-real gaps for the input point clouds, sim-to-real gaps for the generated parameters (the parameter generator depends on estimated object primitives and poses, which performs worse in the real world), and sim-to-real gaps for the interaction dynamics. We did randomize dynamics parameters (mass, friction) during simulation data generation to roughly cover the set of objects that our real robot, the Franka Emika Panda arm, would be capable of manipulating. Anecdotally, we did not observe significant sim-to-real gaps for the precondition model that occurred due to dynamics mismatches. This is partly due to the inherent robustness of our controllers and hardware and partly due to the fact that the real-world objects do not have dynamics that significantly deviate from those of the objects we chose for simulation. Most precondition failures in the real world came from the second source - inaccurate object perception - which led to generating infeasible and out-of-distribution skill parameters that the precondition model has not seen during training.
>
> **Precondition success thresholds.**
> The precondition threshold values are skill-specific and implicitly depend on the end-effector and object geometry. We chose their values to exclude reached poses that have rotated into a different stable pose or translated more than the object’s dimension away from the subgoal pose. However, our method is flexible on the specific thresholds used to label positive and negative preconditions. This is because when the planner uses the precondition model, it can use a prediction threshold other than 0.5, allowing less false positives or less false negatives. These are hyperparameters that can be tuned to optimize downstream task success rate. We did not do so because of the high compute and time requirements (running full suites of planning evaluations and training many precondition models are expensive), but this can be done if needed to improve performance and to adapt the approach to a new domain.
>
> **Related works.**
> We have made significant revisions to our related works section to include more detailed discussions on prior works in nonprehensile manipulation.
>
> **Mathematical rigor.**
> We have updated our paper draft with additional mathematical formulation when introducing the HFVC synthesis optimization problem, and we have added clarifications on planning state and object pose representations. We note that the location of mathematical symbols and equations in the Appendix, as opposed to the main text, was a choice we made in order to satisfy the page limit. Our overall system contains many parts, and it is difficult to cover both the breadths and depths of each part under the page limit. We focused the content in the main text on what is new about our approach, instead of parts that have been explored in the past.